# Analysis of the Polymer Two-Layer Protective Coating Impact on Panda-Type Optical Fiber under Bending

**DOI:** 10.3390/polym14183840

**Published:** 2022-09-14

**Authors:** Yulia I. Lesnikova, Aleksandr N. Trufanov, Anna A. Kamenskikh

**Affiliations:** Department of Computational Mathematics, Mechanics and Biomechanics, Perm National Research Polytechnic University, 614990 Perm, Russia

**Keywords:** polarization-maintaining fiber, Panda fiber, polymer, optical fiber coating, deformation behavior, refractive indices, birefringence, contact, temperature, bending, viscoelastic polymer protective coating

## Abstract

The article discusses the effects of thermal-force on the Panda-type optical fiber. The studies used a wide temperature range. The research used two thermal cycles with exposures to temperatures of 23, 60 and −60 °C. The field of residual stresses in the fiber formed during the drawing process was determined and applied. Panda was considered taking into account a two-layer viscoelastic polymer coating under conditions of tension winding on an aluminum coil in the framework of a contact problem. The paper investigated three variants of coil radius to analyze the effect of bending on fiber behavior. The effect of the coating thickness ratio on the system deformation and optical characteristics was analyzed. Qualitative and quantitative patterns of the effect of temperature, bending, thickness of individual polymer coating layers and relaxation transitions of their materials on the Panda optical fiber deformation and optical characteristics were established. Assessment of approaches to the calculation of optical characteristics (values of the refractive indices and fiber birefringence) are given in the framework of the study. The patterns of deformation and optical behavior of the Panda-type fiber with a protective coating, taking into account the nonlinear behavior of the system materials, were original results.

## 1. Introduction

### 1.1. Research Objectives

The study purpose: to analyze the effect of the viscoelastic polymer protective coating (PC) on the Panda type optical fiber operation over a wide temperature range.

Research objectives:To describe the behavior of PC polymeric materials in terms of viscoelasticity using the generalized Maxwell model.To determine the residual stress fields in an optical fiber during high-temperature drawing from a preform.To create a numerical model of the thermal cycle procedure when loading two stages in the range of –60 to 60 °C followed by exposure to room temperature.To create a parameterized numerical model of an optical fiber wound on an aluminum coil with a constant preload force taking into account the contact interaction.To investigate the influence of the coil radius on the Panda fiber optical characteristics.To study the influence of the thickness ratio of primary and secondary coatings on deformation and optical characteristics.

### 1.2. Problem Context

At present, optical fibers are used in almost all areas of science and technology. They transmit a signal over long distances without loss and at high speed [1] and are used for decorative and landscape lighting [2,3]. Instrumentation based on fiber optic sensors (FOS) is one of the most developing research areas: with applications for temperature [4], displacement [5], pressure [6], bending [7], air quality [8], chemical and biological presence substances [9], angular velocities [10] and other physical parameters. Their universal characteristics contribute to the following: small dimensions and weight, resistance to electromagnetic interference, low power consumption, resistance to aggressive environments, and the ability to work in a wide temperature range.

Many different types of optical fibers are used as sensing elements of FOS exist. Special optical fibers capable of transmitting only one polarized mode of transmitted radiation can be distinguished in this case. It is a polarization-maintaining single mode fiber (PMF). The Panda type PMF is the most popular design for this fiber type. It is used in fiber-optic gyroscopes, coherent communication lines, and in distributed sensors of pressure, temperature, and other physical quantities [11,12,13]. Panda fiber is a heterogeneous structure made of doped quartz glass. A protective coating must be applied to the optical fiber during the drawing process. It protects the fiber surface from contact with dust or moisture particles, which significantly reduces its strength [14,15]. In addition, coatings protect the fiber from chemical and mechanical damage; reduce the effect of micro-bends on its optical characteristics, etc. The coating material determines the possible areas of the fiber practical application. Conventionally, they can be divided into metal [16,17], carbon [16,18] or polymer [16,19,20]. Active development of functional materials for coatings and interlayers can be noted [21,22,23,24]. New materials have improved properties: antifriction [25], corrosion resistance [26], self-healing [26,27,28], etc. But the possibility of their application in fiber optics requires large-scale research. Acrylates, silicones, and UV-cured polymers are main materials of polymer coatings in the fiber industry. Each has its own areas of application and operation temperature range. The protective coating is one or more layers of polymers selected in such a way as to solve the above problems most effectively [16,29]. A two-layer UV-curable polymer coating is used on the fiber under study.

There are several studies related to the assessment of the influence of geometric characteristics [30], physico-mechanical and operational properties of quartz glasses [31], production process parameters [32], etc. Almost all studies related to determination of the fiber deformation and optical characteristics are considered within the framework of the geometric configuration significant simplifications: without section geometry using averaged properties, without protective coating [33,34,35], in the framework of the thermoelasticity theory, without relaxation processes. Simplification of the object under study models leads to inaccuracies and significantly affects the qualitative and quantitative level on the obtained patterns of changes in optical and deformation characteristics within the framework of technological and operational processes. This significantly limits the effectiveness of development in this area.

### 1.3. Problem Description

An important research area is the analysis of effects of protective coatings on crack resistance, strength, destruction, and operation of optical fibers [36,37,38,39]. Many studies are aimed at rationalizing fibers operation and expanding the scope of their application [39,40]. A two-layer PC with a primary coating of DeSolite 3471-1-152A and the secondary coating of DeSolite DS-2015 is one of the options for protecting Panda optical fibers. Relaxation transitions in DeSolite 3471-1-152A and DeSolite DS-2015 materials in the considered temperature range are revealed in [41]. They contribute to the stress-strain state of the structure. It is of interest to model the behavior of the Panda-type fiber taking into account polymer coatings at operating temperatures of −60 to +60 °C. Optical fibers interact with each other, with metal and non-metal elements as part of the production technological process, rejection of finished products, in real structures and systems. The optical fiber study taking into account the mechanics of contact interaction is also relevant.

External and internal factors affect fiber performance: load [20,42], bending [20,42,43], temperature [44], thermal shrinkage [33], mating with different surfaces [45], etc. Furthermore, Trufanov et al. [46] considered the influence of structural elements on deviations from design parameters. Miniaturization of optical fibers required for industrial, chemical, and biological probing in inaccessible places is of interest [47,48]. This leads to the need for creating efficient models of optical fibers and their study in the framework of computer engineering. Analysis of the influence of fiber material properties and polymer coatings, the structure geometry, external influences, the adhesion of elements, and the contact interface of waveguides with different objects will provide important data on the product operation. Research data can be used to rationalize the design and expand the scope of optical fibers.

This work is aimed at modeling Panda-type optical fiber, taking into account the heterogeneous properties of the structural elements, polymer coating and contact interaction with an aluminum coil as part of a technological test in a wide temperature range from −60 to +60 °C. The influence of the thickness ratio of the two-layer polymer coating on optical fiber operation is also considered by the study.

## 2. Materials and Methods

### 2.1. Model

The fiber that was considered had an ideal geometry (Figure 1). The fiber diameter was 80 µm. The fiber was designed to transmit the signal at wavelength 1.55 µm. The light-conducting core diameter was 6 µm. The power rod diameter was 15 µm. The distance between the fiber center and the power rod was 15 µm. The secondary PC diameter was 167 μm. The primary PC diameter was 130 μm (taken as a standard diameter). The PC thickness h was 43.5 μm. The standard ratio of the primary and secondary layers thicknesses was 42.5 and 57.5 % respectively, of the multilayer coating total thickness.

A fiber with tension force of 0.2 N was wound in one row on an aluminum coil and subjected to thermal cycling according to the given law within the framework of the experiment. The calculation scheme with the PC thicknesses standard ratio is shown in Figure 1.

The research considered seven ratios of the thicknesses of the primary and secondary layers while maintaining the overall size (Table 1). The analysis of the effects of the PC thicknesses ratio on the fiber deformation and optical characteristics was the aim of the study.

The fiber was wound on an aluminum coil as part of the test. Standard coil radii were 22 and 50 mm. The standard coil radii were much larger than the characteristic fiber size. Therefore, the effect of bending, tightness, and other factors was insignificant. The decision to coil the radius to 5 mm was taken to better illustrate the effect.

### 2.2. Materials

The Panda-type optical fiber is a construction of quartz glasses and a two-layer UV-curable polymer coating (Figure 1). The calculation scheme included seven materials. The light-conducting core consisted of quartz doped with germanium oxide GeO_2_ (mat. 1). Doping alters the refractive index, which creates conditions for total internal reflection of light at the core-fiber interface. The power rod consisted of two layers. The inner layer was quartz doped with boron oxide (mat. 2). The outer layer was quartz doped with oxides of boron B_2_O_3_ and phosphorus P_2_O_5_ (mat. 3). Additives significantly increase the thermal expansion coefficient, reduce viscosity and glass transition temperature. This makes it possible to form the necessary stress field in the core during the fiber manufacturing process, which determines its optical characteristics. The main fiber material was quartz glass (mat. 4). Low modulus polymer DeSolite 3471-1-152A was used as the primary PC (mat. 5). It is in a highly elastic state at room temperature. Its key task is reducing the influence of external forces and micro-bending on the fiber. DeSolite DS-2015 polymer was applied as a secondary PC (mat. 6). Its task is to protect against mechanical damage. The coil was made of aluminum (mat. 7). The elastic properties and thermal expansion coefficients (TEC) of the materials are presented in Table 2. The properties of the fiber materials were obtained experimentally. Free compression modules and Poisson’s ratios of materials 1–3 were calculated using mixture formulas. Thermo-mechanical properties of the aluminum coil were taken from reference literature.

Suprasil 300 (Heraeus Quarzglas GmbH & Co. KG, Hanau, Germany) is the main material for the manufacture of optical fiber [49]. Doping of quartz to create the necessary properties of the light-conducting core and power rods occurs during fiber production. Protective coatings DeSolite 3471-1-152A [50] and DeSolite DS-2015 [51] are manufactured by DSM Desotech Inc. (Elgin, IL, USA).

The relationships of the generalized Maxwell model with the description of the relaxation functions by the Prony series were used in the numerical simulation of the behavior of PC materials (Figure 2) [52].

The temperature-time analogy described by the Williams–Landell–Ferry (WLF) equation was used to account for the effect of temperature on the polymer coatings properties [53]. For the WLF equation, the following parameters were chosen for the primary coating: the reference temperature was 0 °C, the material constants were c_1_ = 20.036 and c_2_ = 32.666, the relaxation function was described by the sum of 18 members of the Prony series. For the secondary coating: the reference temperature was −70 °C, the material constants were c_1_ = 40242.2827 and c_2_ = 267448.888, and there were 60 terms in the Prony series. The search for the model’s parameters was carried out by the Nelder–Mead method based on the experimental data.

The doped silica glasses behavior was described in terms of a Maxwell-type linear viscoelasticity model with temperature-dependent viscosity to simulate residual stresses in an optical fiber.

### 2.3. Thermal Cycle and Prestressing

The study was carried out within a thermal cycle framework considering the residual stresses in the optical fiber (Figure 3). Two cycles of loading in the range of −60 to +60 °C with holding at room temperature after the second cycle were used to reveal the relaxation processes taking place in the PC. The hypothesis about a uniform temperature change throughout the volume of the fiber and coil was accepted in the simulation. The rate of temperature change in the thermal cycle was low.

Six holding zones at a constant temperature were allocated for the analysis of the thermal cycle: I was 23 °C; II was 60 °C; III was −60 °C; IV was 60 °C; V was −60 °C; VI was 23 °C. Deformation and optical parameters were considered in the middle excerpt time interval.

Initial elastic deformations were set in each node of the fiber finite element model before the calculation starts to take into account the prestressed state.

Residual deformations were determined during preliminary simulation of fiber cooling during high-temperature drawing from 2000 °C to room temperature. The calculation was carried out according to the method described in [46] which includes the sequential solution of the nonstationary heat conduction boundary value problem and the boundary value problem of changing stress fields. Residual stress fields that create the necessary conditions for the polarization of light propagating through the fiber were formed after drawing (Figure 3b). The residual stress level was quite high. Tensile stresses along the fiber in power rods reached the highest values.

Flexural strains were calculated using the planar section hypothesis. Deformations equal to the distance from the neutral section to the node divided by the sum of the coil and fiber radii in PC were specified at each node. Tensile strains were calculated according to Hooke’s law considering fiber cross-section heterogeneity. They were the same for all nodes of fiber and PC.

### 2.4. Numerical Finite Element and Methods

The numerical implementation was performed by the finite element method in the ANSYS Mechanical APDL (ANSYS Inc., Canonsburg, PA, USA) software in a three-dimensional formulation. SOLID185 quadrilateral eight-node finite elements with Lagrangian approximation were used. The finite element model was built taking into account the optical fiber section geometry. The research assessed reducing the finite element size in the light-conducting core and near the contact zone. The finite element maximum volume of the fiber was 506 µm^3^. The finite element minimum volume in the core was 3.5 µm^3^ and near the contact area was 57 µm^3^. The degree of model discretization was 57 thousand nodal unknowns (53 thousand in the optical fiber volume with PC). The solution was carried out according to the axial symmetry scheme. Only the sector of design with a central angle of 0.5° was studied.

The bending and tightness of the fiber on the coil, considering the viscoelastic properties of PC materials, can cause uneven contact pressure distribution. Modeling the problem in a simplified axisymmetric formulation can distort the results. The axisymmetric geometry makes it possible to consider the construction sector and track the contact pressure non-uniformity. The decision to consider the sector of the fiber-reel system near the central section was made within the framework of the first approximation. This allows you to track the effect of load on the contact zone. The model was parameterized relative to the central angle. The study of the influence of the central angle value on system deformation is possible and will be carried out in further studies.

The fiber-coil conjugation was modeled using the elements contact pair CONTA173 and TARGE170. “Close gap” was the initial setting of contact elements. The augmented Lagrangian method is a contact computation method. Friction is not taken into account in the model. However, the approach described above allows it to be taken into account in the future. The contact boundary conditions include sliding without friction (1) and no contact (2):(1)$${\vec{u}}^{1}\neq {\vec{u}}^{2},\;\sigma_{n}^{1}=\sigma_{n}^{2},\;\sigma_{n\tau }^{1}\neq \sigma_{n\tau }^{2},$$
(2)$$\left|u_{n}^{1}-u_{n}^{2}\right|\geq 0,\;\sigma_{n\tau_{1}}=\sigma_{n\tau_{2}}=\sigma_{n}=0,$$
where $\vec{u}$ is contact boundary displacement vector; $u_{n}$ is normal displacements; $\sigma_{n}$ is stress along the normal to the contact boundary; $\sigma_{n\tau_{1}}$, $\sigma_{n\tau_{2}}$ are tangential stresses at the contact boundary; $\sigma_{n\tau }$ is the value of the tangential contact stresses vector; 1–2 are conditional numbers of contact bodies.

### 2.5. Optical Characteristics of the Fiber

Refractive indices, modal and material birefringence are the main optical characteristics of waveguides. They arise because of asymmetric residual stresses around the light-conducting core, initiated by power rods. These indicators make it possible to evaluate the system parameters influencing the conditions for the signal to pass through the light-conducting core. A fairly large number of options for calculating fiber optical characteristics through the stress-strain state parameters exist [54,55,56]. Comparison of the optical characteristics obtained by the formulas described in the above sources was carried out as part of the work. It was established that the values of the obtained parameters differ by no more than 5%.

The stresses in the light-conducting core and the refractive indices are related by the Equations (3) [57]:(3)$$\Delta n_{x}=-C_{1}\sigma_{x}-C_{2}\left(\sigma_{y}-\sigma_{z}\right),\;\Delta n_{y}=-C_{1}\sigma_{y}-C_{2}\left(\sigma_{x}-\sigma_{z}\right),$$
where $\sigma_{x}$, $\sigma_{y}$, $\sigma_{z}$ are stress tensor components; $C_{1}=-6.5\times 10^{-13}$ Pa^−1^, $C_{2}=-4.22\times 10^{-12}$ Pa^−1^ are photoelasticity coefficients.

The changes in the refractive indices from the light-conducting core center are also optical characteristics. They are calculated by the Formula (4):(4)$$\Delta {\tilde{n}}_{x}=\Delta n_{x}-{\Delta n_{x}|}_{\vec{x}=\left(0;0;0\right)},\;\Delta {\tilde{n}}_{y}=\Delta n_{y}-{\Delta n_{y}|}_{\vec{x}=\left(0;0;0\right)},$$
where ${\Delta n_{x}|}_{\vec{x}=\left(0;0;0\right)}$, ${\Delta n_{y}|}_{\vec{x}=\left(0;0;0\right)}$ are refractive indices at the core center.

Modal and material birefringence is calculated by Formulas (5) and (6), respectively:(5)$$B=\left(\left(C_{1}-C_{2}\right)\int_{S_{c}}\left(\sigma_{x}-\sigma_{y}\right){\left|E^{*}\right|}^{2}dS_{c}\right)/\left(\int_{S_{c}}{\left|E^{*}\right|}^{2}dS_{c}\right),$$
(6)$$B_{m}=\left(C_{1}-C_{2}\right)\left(\sigma_{x}-\sigma_{y}\right),$$
where $E^{*}$ is fundamental mode intensity distribution over the fiber cross-section, $S_{c}$ is core cross-sectional area.

## 3. Results

### 3.1. The Analysis of the Influence of the Coil Radius on the Fiber Performance under Bending and Tightness

The fiber prestressed state is made up of residual stresses, bending stresses and tension. It is necessary to make an assessment of the influence of the coil radius on the system deformation and optical characteristics. The coil radius must be chosen to enhance the effect of bending and tightness on the system behavior. At the first stage, the research studied a standard thickness ratio PC for three coil radii of 5, 22, and 50 mm.

The coil radius and fiber bending have the greatest influence on the stress tensor component $\sigma_{z}$ and the contact pressure (Figure 4). The stress level $\sigma_{x}$ and $\sigma_{y}$ at contact and bending changes insignificantly (less than 1%).

Fiber bending had a significant effect on the system stress-strain state for the coil with a radius of 5 mm. The pattern of the stress distribution over the fiber cross-section was asymmetric. The maximum stress level for the $\sigma_{z}$ component increased by more than 2.5 times from 205 to 527 MPa compared to the residual stresses. The maximum $\sigma_{z}$ level increased by only 48.6 and 33.1% compared to the residual stresses at coil radii of 22 and 50 mm, respectively.

An increase in the contact pressure maximum level by 4.44 and 1.39 times was observed at $R_{\mathrm{coil}}$ equal to 5 and 22 mm, respectively, relative to a 50 mm coil. The contact pressure maximum level was observed during the first heating up to +60 °C. The coil radius also affected the temperature at which the contact surfaces were completely detached due to assembly materials thermal shrinkage. Contact opening occurred at a temperature of approximately 17 °C with a coil radius of 5 mm, at 22 and 50 mm at −3 °C.

Changes in the refractive indices in the two orthogonal axes of light propagation in the light-conducting core center and birefringence are shown in Figure 5.

Nonlinear changes in the refractive indices were observed with a 5 mm coil. This was due to the bending and relaxation transitions in PC. This effect was not observed with large coils. The pattern of the change in birefringence within the framework of a thermal cycle does not depend on the coil radius. The B quantitative values at the 5 mm coil were no more than 2.2% less than those with large coils.

Optical parameters with 22 mm and 50 mm coils had small differences. The main differences were observed in the holding zones at a maximum positive temperature of 60 °C (no more than 2.5%) and at the end of the thermal cycle in the holding zone at room temperature (no more than 6.74%).

The material birefringence isofields in a light-conducting core in contact with different coils are shown in Figure 6.

The pattern of the material birefringence distribution also depended little on the coil radius. The influence of the coil radius on the minimum and maximum level of the parameter did not exceed 2%.

The effect of bending, tightness, contact and PC viscoelastic properties on the parameters of the stress-strain state and the optical characteristics of the fiber was better seen on a coil with radius of 5 mm. The decision was taken to conduct all further studies at $R_{\mathrm{coil}}=5$ mm. Reducing the coil did not distort the optical signal but allowed revealing the influence of material properties and system parameters on the performance of the system.

### 3.2. Analysis of the Influence of PC Thickness Ratio on Deformation and Optical Characteristics

The research analyzed the effect of the PC thickness ratio on the stress-strain state and optical characteristics of the fiber in the second stage of the study.

The change in the stress tensor components in the core center within the framework of the thermal cycle is shown in Figure 7.

The negative temperature had the greatest influence on the pattern of the change in the $\sigma_{x}$ and $\sigma_{y}$ components. This effect was associated with relaxation transitions in a low modulus buffer layer. At −60 °C the primary layer of the protective coating was in a vitrified state. The spread of $\sigma_{x}$–$\sigma_{y}$ values at −60 °C did not exceed 10–20 MPa. The maximum values were observed at the standard PC ratio. The dependence $\sigma_{x}$–$\sigma_{y}$ on the PC thickness ratio was linear at positive temperatures.

The $\sigma_{z}$ makes the maximum contribution to the stress tensor. Bending, fiber tension and relaxation transitions of the secondary PC had a significant effect on $\sigma_{z}$. The non-linearity of the parameter change were noted in the holding zones at 60 °C. The greatest deviations of the parameters occurred in the first section of heating up to 60 °C. The values spread was 78–135 MPa. An increase in $\sigma_{z}$ was observed with an increase in the thickness of the primary PC in the holding areas at −60°C. The pattern of the change was linear. A decrease in $\sigma_{z}$ was observed with an increase in the thickness of the primary coating in the zones of holding at positive temperatures. Dependencies were non-linear. Two loading cycles led to a decrease in $\sigma_{z}$ at room temperature by an average of 25%.

Contact pressure is the main characteristic of the contact zone without friction. Figure 8 shows the dependence of the maximum contact pressure in the thermal cycle. The maximum contact pressure was observed at the initial contact point of the fiber-coil for all options of the thicknesses ratio of the primary and secondary coatings.

Relaxation transitions in materials affect not only the system stress-strain state, but also the contact pressure. This is due to the continuous change in the contact interaction area under the influence of temperature. A slight influence of $h_{2}$ on the qualitative and quantitative patterns of changes in the contact zone parameters was observed when the secondary PC thickness was more than 50% of h. The secondary PC material perceived force loading to a lesser extent. The contact pressure was distributed over a smaller area than with a standard PC thickness distribution.

A significant effect of temperature and $h_{1}$ on the multilayer coating deformation was observed when the primary PC thickness was more than 50% of h. This led to a more uneven change $\max P_{K}$: significant decrease zones in the parameter appear, the contact pressure tends to 0 in an unfavorable case. This effect was associated with a significant deformation of the primary PC and the contact interaction area with a slight change in the force loading.

The divergence of contact surfaces for PC thicknesses at different ratios occurs at different temperatures. Full opening of the contact for PC thickness ratios of all variants was observed in the temperature range of 2–19 °C. This is due to system materials thermal shrinkage. The maximum divergence of mating surfaces was observed at −60 °C. For example, at the standard thickness ratio PC was 8.3 µm.

The change in refractive indices within the framework of a thermal cycle is shown in Figure 9.

Thermal cycling results in a decrease in the refractive indices of the core relative to the fiber in the quiescent state. The *y*-axis index decreases more than the *x*-axis. Polymeric materials relaxation transitions have the maximum effect on the first loading cycle.

The sign change of $\Delta n_{x}$ occurred at thick secondary PC at the temperature below –24 °C. This is due to contact loss with the coil, the large effect of bending and thermal deformation, and the phase transition in the primary PC. A thin primary buffer layer ceases to fulfill its protective functions.

The stress $\sigma_{z}$ had the greatest influence on the parameters. The difference in refractive indices tended to be minimal with an increase in the primary coating thickness at all temperature options. The smallest scatter of values was observed at primary PC thickness of more than 70%. The refractive index changed within $2\cdot 10^{-4}$. The thick inner sheath smoothed out the effect of temperature on the change in the refractive indices in the light-conducting core.

The change in the refractive indices from the light-conducting core center is considered in Figure 10. The main effects were observed on the first loading cycle. The change in parameters after the complete loading cycle at room temperature was also of interest. Changes in parameters at stages I–III and VI were therefore considered.

Thermal cycling had an insignificant effect on the level and pattern of the change in refractive indices with distance from the core center. The indicator decreased at 60 °C, at –60 °C it increased. Large changes in the refractive index occurred along the slow axis. Tightness, bending and the pattern of the conjugation significantly affected the changes in the refractive indices with distance from the core center. Qualitatively, the refractive index profile corresponds to the cases described in the literature [58,59].

Figure 11 shows the change in the modal birefringence over the thermal cycle holding zones. The modal birefringence of the core quantitatively depended on the environment temperature. The parameter level was quantitatively similar to the data in Li et al. [55]. The research established quantitative parameters of birefringence fluctuations with temperature change.

The birefringence of the core increased with a magnification in the primary layer PC thickness at $h_{1}/h_{2}>h_{\mathrm{standart}}$ and a temperature of −60 °C. A decrease in B was observed when $h_{1}$ decreased to 30% of the total PC thickness and then an increase in mode birefringence was observed. The B increased by more than 0.3%.

The core birefringence increased with magnification in the primary layer PC thickness at the temperature of 60 °C. The B maximum value was observed at $h_{1}$ 90%. The B was 0.1% larger than at the standard thickness ratio PC.

Two loading cycles in the −60 to +60 °C range have the maximum effect on fiber birefringence at the room temperature of 23 °C. The pattern of the B change from the thickness PC was switched to the opposite after the second loading cycle. The B maximum value was observed at $h_{1}$ 90%. The B was 0.14% larger than at the standard thickness ratio PC. The PC thickness had little effect on the modal birefringence overall.

It can be concluded that a change in the PC thickness ratio does not have a strong distortion on the optical characteristics. However, it allows you to change the protective properties of the coating. It was established that the primary PC optimal thickness was in the range of 30 to 70% of the total PC thickness. An additional study of the effect of PC thickness on the waveguide near the standard PC thickness ratio is required.

## 4. Discussion

### 4.1. Limitation Statement

This paper presents numerical simulation results of the behavior of the Panda-type optical fiber over a wide range of temperature change and bending. The model has several limitations:Frictionless contact on the fiber-coil interface was modeled.The surfaces joint deformation of the protective coatings and the quartz fiber was considered.A constant TEC of materials was used in the model. In reference [41] the dependence of the protective coating materials TEC on temperature was established.The fiber has an ideal cross-sectional geometry.Uniform temperature change throughout the fiber and coil volume with its small change in the thermal cycle.

Further directions of work development:Accounting for friction on the interface fiber-coil and introduction of contact between the PC layers and the quartz base is needed. The analysis of the influence of adhesion on system behavior is required.It is necessary to consider the dependence of PC materials TEC on temperature in the model. Materials models with variable TEC are built. Preliminary studies are completed.The analysis of the influence of temperature with different rates of its change both for different thermal cycles and within the same thermal cycle should be performed. Interest in this type of research is noted in [60].Rationalization and reduction of the PC thicknesses and its layers thicknesses ratio to minimize the fiber overall dimensions without losing the product deformation and optical characteristics is of interest.

### 4.2. About Birefringence

Internal birefringence, caused by stress in polarization-maintaining optical fibers, is one of the most important characteristics. Birefringence occurs due to the stress difference between the two principal axes. This effect occurs due to the power rods, which have a TEC different from the base material.

The numerically found residual stresses fields are used to calculate birefringence in our work. There are many works in which a birefringence analytical estimate for a Panda-type fiber has been proposed [34,54,55]. Liu et al. [34] evaluated previously published formulas for calculating birefringence, including those for asymmetric fiber geometry. Analytical formula for birefringence calculating of the Panda-type fiber is derived from the study by Liu et al.:(7)$$B=\left(-4\cdot C\left(\lambda \right)\cdot E\cdot \Delta \alpha_{s}\cdot \Delta T_{s}\right)/\left(1-v\right)\cdot {\left(r/d\right)}^{2},$$
where C(λ)=3.36e−12 Pa^−1^ is photo-elastic constant for wavelength λ=1.55 nm, E=72000 MPa is Young’s modulus mat. 4, v=0.17 is Poisson’s ratio mat. 4, r=7.5 μm is power rod radius, d=15 μm is distance between centers of power rod and fiber, $\Delta T_{s}=-700$ K is difference between the materials softening temperature of the power rod and the quartz shell, $\Delta \alpha_{s}$ is difference in TEC between the materials of the power rod and the quartz shell.

$\Delta \alpha_{s}$ is calculated by Formula (8) and is equal to 2.365 × 10^−6^ K^−1^:(8)$$\Delta \alpha_{s}=\alpha_{SAP}-\alpha_{mat.4},$$
where $\alpha_{SAP}$ is averaged TEC of the power rod material calculated according to the materials 2 and 3 volume fraction, $\alpha_{mat.4}$ is TEC of material 4.

The B values obtained in our study were comparable with the analytical values calculated by Formula (7). The birefringence values for the thermal cycle VI section for fiber with the thicknesses PC standard ratio are presented in Table 3. Some coefficients were taken from [34] in the analytical solution, which introduces an additional error. Δ is relative difference between the analytical and numerical birefringence in %, where the analytical solution is taken as the reference value.

The discrepancy between the analytical solution and the numerical simulation results did not exceed 5%. The analytical formula was based only on fiber geometry and material properties. The softening temperature $T_{s}$ is not constant and depends on the glass composition and the cooling rate. The birefringence obtained in the numerical solution framework considers the external conditions that affect the internal and external fluctuations of the waveguide parameters along the fiber length. In our study, these are bending, thermal cycling and contact with the coil. The materials properties introduce an additional error into the analytical calculation. The pure materials properties are more or less studied. But the product includes glasses with different alloying additives and their percentages. Finding properties of such glasses is a difficult task. Therefore, estimates can only be approximate. Double-layer power rods were used in the fiber. The literature considers power rods of only one material SiO_2_–B_2_O_3_. The TEC average value for the materials of power rods was entered into Formula (8) in this way. This fact can also introduce an additional error into the analytical solution. The birefringence value of the Panda fiber varies within $\left[2÷5\right]\cdot 10^{-4}$ in different works [56,61,62]. The values and dependences of fiber birefringence obtained by us were in good quantitative and qualitative agreement with the other authors’ results. This fact confirms the correct operation of the Panda-type fiber model. The model can be used for further analysis of product performance.

### 4.3. Main Results

The creation of a three-dimensional model of the polarization-maintaining single mode fiber behavior taking into account many internal and external factors that affect the passage of light through the fiber was the main result of the study. The influence of parameters one or two on the fiber operation is most often found in the works: temperature, external loads, radiation, bending, wavelength, change in the geometry of structural elements, etc. For example, Lesiak and Woliński [33] considered the effect of polymerization shrinkage on the composite material in which Panda fiber was embedded without PC. They noted that such a process changes the initial stress distribution within the fiber due to its deformation. Li et al. [56] considered effect of changing the geometry of a light-conducting core on birefringence. All the above studies considered problems in a plane elastic formulation.

The effect of bending in the range of 2–60 mm radii and temperature in the range of 60 °C with a temperature change rate of −40 to +65 °C/min on the Panda fiber birefringence was considered by Wang H. et al. [63]. The conclusion that bending at a radius of more than 10 mm practically does not affect the change in birefringence was made. The change magnitude in birefringence was calculated using an analytical formula that did not take into account all the fiber operating conditions. Birefringence induced only by bending increases with decreasing bending radius. Birefringence decreased as the bending radius decreased in our study. A decrease in birefringence magnitude with increasing temperature was noted. This effect is also seen in our study in Figure 11a.

Residual stresses after cooling the fiber from 2000 °C to room temperature after high-temperature drawing from the preform were found for the correct operation of the model. The maximum level of residual elastic stress in the fiber after drawing varies from approximately 150 MPa according to the Yablon research data [62]. Within the framework of the model implemented in this study, the maximum level of residual stresses reaches: $\left|\max\sigma_{x}\right|=148.5$ MPa; $\left|\max\sigma_{y}\right|=181.5$ MPa; $\left|\max\sigma_{z}\right|=205.5$ MPa. This fact reflects the constructed model’s performance and the possibility of their use for a wide range of the tasks.

Lack of information about behavior models and thermomechanical properties of materials and their elements affects the presented results reliability. Searching for information from open sources about glasses is quite labor-intensive. The open literature gives the limited data on materials properties. Therefore, a large-scale study of the influence of dopants and their percentage on optical fiber structural elements operation is of interest. Viscoelastic models of the behavior of protective coating materials were developed in the presented research. This makes a significant contribution to the technological mechanics of the polarization-maintaining single mode fiber.

The modern world trends follow the path of miniaturization of measuring instruments. The development of micro/nano devices with low manufacturing cost and energy consumption but high accuracy is one of the priority areas [64,65]. There are many areas of application [66,67]: portable navigation devices, space satellites and probes, unmanned devices, etc. Large-scale and conventional instruments cannot be used there. Weight and dimensions are one of the most important indicators.

We are faced with more complex mathematical models, nonlinearities, and the need to consider the materials viscoelastic properties with a decrease in size [68]. Simplified models of products and materials do not allow describing the structure’s behavior. Modern possibilities of computer technology and software allow this to be realized. A digital analog of the fiber under study can be obtained as a result.

Our study confirms that the consideration of the viscoelastic properties of PC materials manifests itself at the micro level for a coil with a radius of 5 mm. The nonlinear behavior of the presented parameters can be traced in Figure 7a, Figure 8 and Figure 9a. This effect was not observed on larger radius coils. Further research will be aimed at rationalizing fiber design to reduce protective coatings diameters.

New dependences of the deformation and optical characteristics under conditions of a complex stress-strain state, considering the viscoelastic behavior of the PC materials, bending along the radii of 50, 22 and 5 mm, tightness and contact with the coil were obtained in the framework of the study.

## 5. Conclusions

Mathematical description of the thermomechanical deformation of Panda-type optical fiber during bending testing after fabrication was done as part of the work. The studies were carried out taking into account polymer multilayer protective coatings. The polymer coating materials behavior was described in terms of viscoelasticity theory. The problem was considered taking into account the contact of the fiber with the aluminum surface without friction at the two-stage thermal cycle.

Qualitative and quantitative regularities were established within the framework of numerical experiments series:
-The change in temperature during the thermal cycle affected the light-conducting core stress-strain state for all considered ratios of the polymer coating thickness. The $\sigma_{z}$ component changed over the entire temperature range, while the $\sigma_{x}$ and $\sigma_{y}$ components changed at negative temperatures.-A more nonlinear pattern of the deformation behavior was observed with an increase in the percentage of the protective coating primary layer.-Permanent deformation and change in the optical characteristics of the fiber occurred when it operated under conditions of large temperature amplitudes. This may affect the quality of the signal.-No contact of the fiber with the coil occurred at different temperatures in the range of 2–19 °C. This was due to the material’s thermomechanical properties and the coating thicknesses percentage.-Primary coating thickness in the range of 30 to 70% of the total thickness was optimal.-The analysis of the existing formulas for determining the fiber optical characteristics was completed. It was established that the difference between the quantitative values of the optical characteristics calculated using different formulas did not exceed 5%.

The scientific novelty of the research results lies in considering the viscoelastic properties of the polymer coating materials, considering the temperature cyclic change, which allowed us to trace the process physics, as well as consider the model in the framework of a three-dimensional formulation when implementing thermo-mechanical loading. The study of the influence of protective coating thicknesses ratio on the structure stress-strain state helped to understand the influence of the material properties on the fiber overall performance.

## Figures and Tables

**Figure 1 polymers-14-03840-f001:**
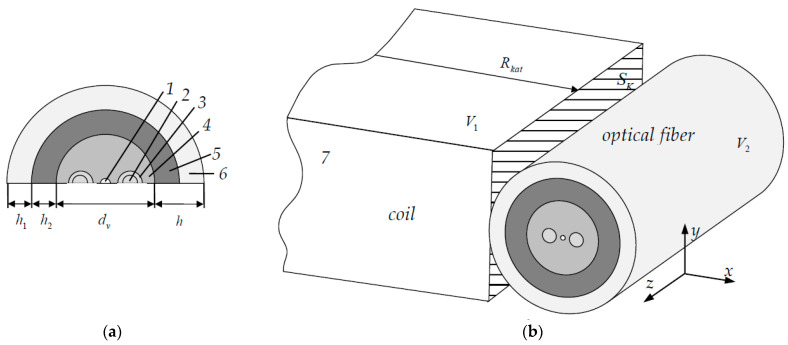
Design scheme: (**a**) the fiber cross-section; (**b**) the three-dimensional model; 1, the light-conducting core; 2, the inner layer of the power element; 3, the outer layer of the power element; 4, the quartz glass; 5, the primary PC; 6, the secondary PC; 7, the aluminum coil.

**Figure 2 polymers-14-03840-f002:**
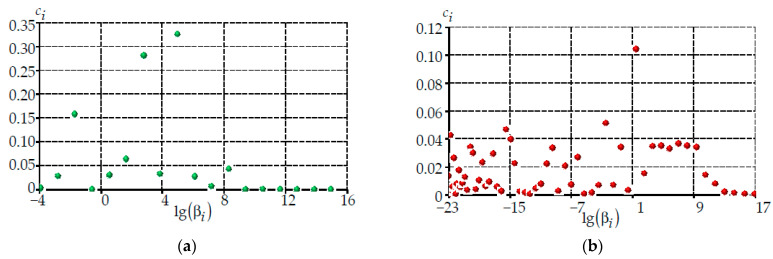
Prony series for polymeric protective coatings: (**a**) is material 5; (**b**) is material 6.

**Figure 3 polymers-14-03840-f003:**
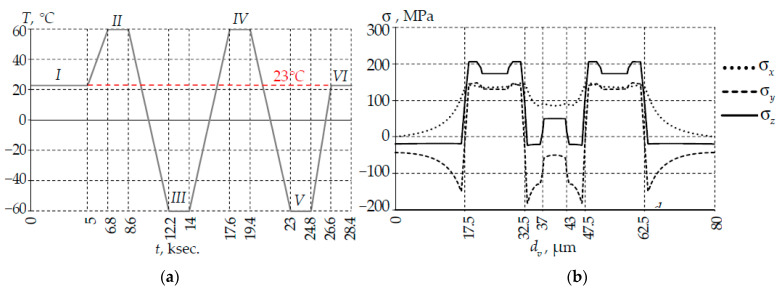
Temperature cycle (**a**) and diagrams of residual stresses along the fiber central section after drawing and polarization (**b**): I–VI are numbers of holding zone at constant temperature; I is 23 °C; II is 60 °C; III is −60 °C; IV is 60 °C; V is −60 °C; VI is 23 °C.

**Figure 4 polymers-14-03840-f004:**
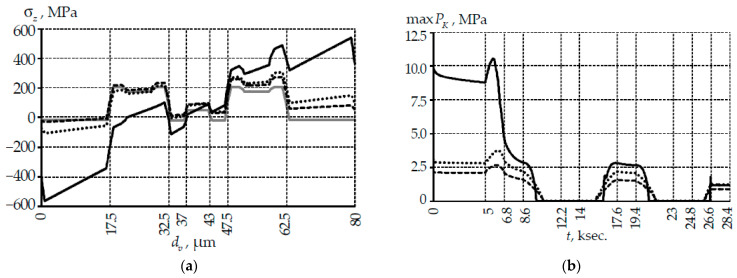
Influence of coil radius and fiber bending on the system deformation parameters: (**a**) is $\sigma_{z}$ along the fiber central section; (**b**) is $\max P_{K}$; the gray line is the residual stress after drawing and polarization; black lines are parameters after bending and tension fiber on the coil; solid line is $R_{\mathrm{coil}}=5$ mm; points is $R_{\mathrm{coil}}=22$ mm; dotted line is $R_{\mathrm{coil}}=50$ mm.

**Figure 5 polymers-14-03840-f005:**
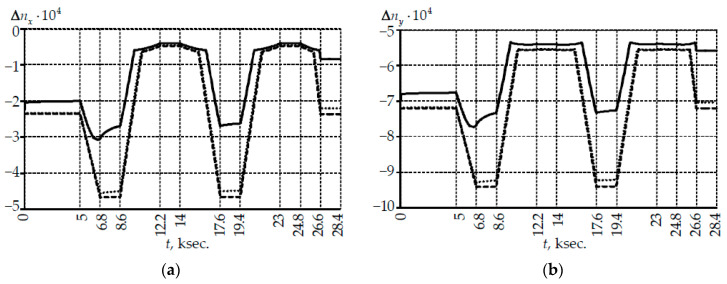
Influence of coil radius and bending on the optical parameters of fiber: (**a**,**b**) are deviations of the refractive indices in the light-conducting core center along the x and y axes, respectively; (**c**) is modal birefringence; solid line is $R_{\mathrm{coil}}=5$ mm; points is $R_{\mathrm{coil}}=22$ mm; dotted line is $R_{\mathrm{coil}}=50$ mm.

**Figure 6 polymers-14-03840-f006:**

Material birefringence: (**a**) where $R_{\mathrm{coil}}=5$ mm; (**b**) where $R_{\mathrm{coil}}=22$ mm; (**c**) where $R_{\mathrm{coil}}=50$ mm.

**Figure 7 polymers-14-03840-f007:**
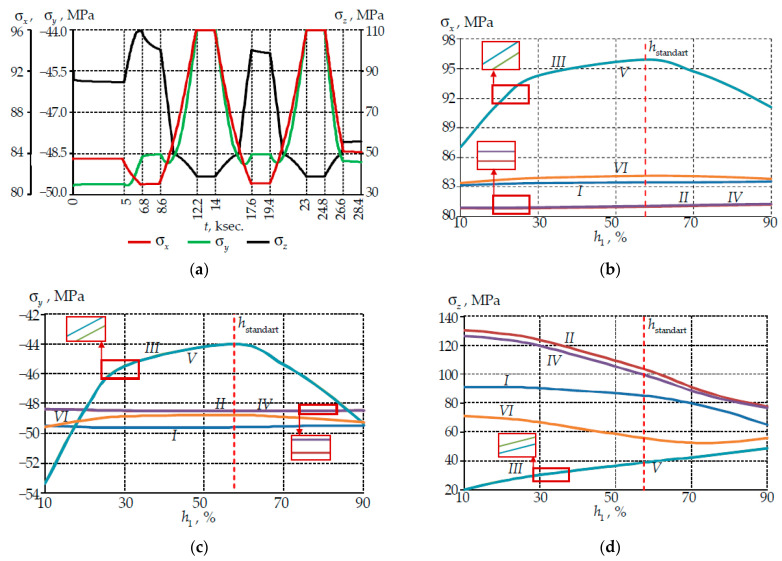
Stresses in the fiber light-conducting core center: (**a**) evolution of stresses within the thermal cycle at standard ratio PC; (**b**) dependence of $\sigma_{x}$ on the PC thickness; (**c**) dependence of $\sigma_{y}$ on the PC thickness; (**d**) dependence of $\sigma_{z}$ on the PC thickness; I–VI are numbers of holding zone at constant temperature; I is 23 °C; II is 60 °C; III is −60 °C; IV is 60 °C; V is −60 °C; VI is 23 °C.

**Figure 8 polymers-14-03840-f008:**
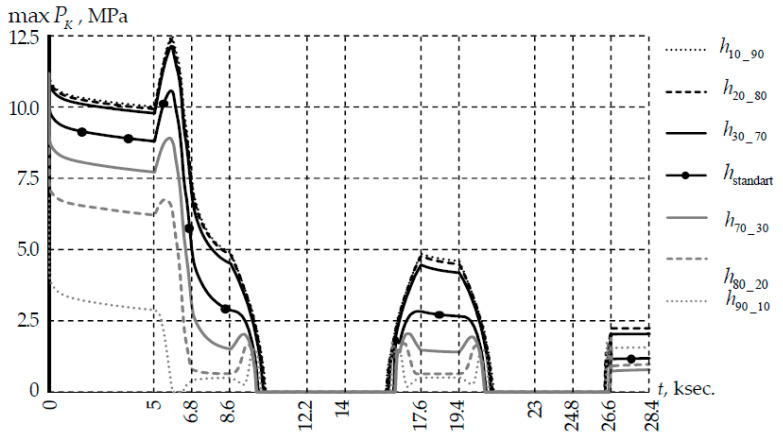
Evolution of contact pressure under thermal cycle conditions.

**Figure 9 polymers-14-03840-f009:**
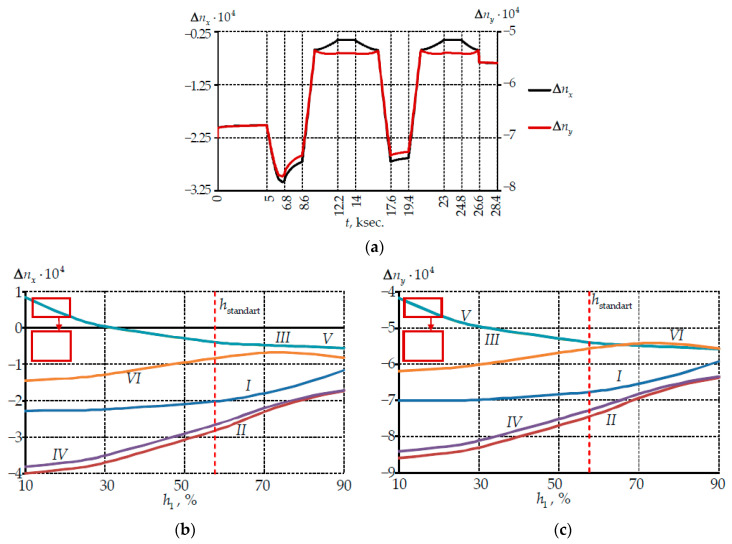
Deviations of the refractive indexes in the light-conducting core center: (**a**) evolution of parameters within the thermal cycle; (**b**) dependence of $\Delta n_{x}$ on the PC thickness; (**c**) dependence of $\Delta n_{y}$ on the PC thickness; I–VI are numbers of holding zone at constant temperature; I is 23 °C; II is 60 °C; III is −60 °C; IV is 60 °C; V is −60 °C; VI is 23 °C.

**Figure 10 polymers-14-03840-f010:**
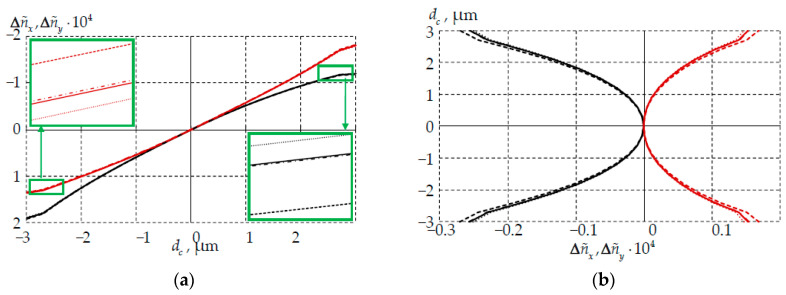
Change in refractive indices from the light-conductor core center: (**a**) slow axis; (**b**) fast axis; dash-dotted line is I zone of the thermal cycle; points is II zone; dotted line is III zone; solid line is VI; black is $\Delta {\tilde{n}}_{x}$; red is $\Delta {\tilde{n}}_{y}$.

**Figure 11 polymers-14-03840-f011:**
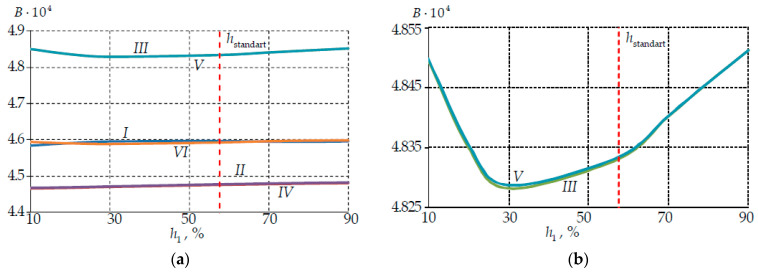
Dependence of modal birefringence on the PC thickness: (**a**) all thermal cycle holding zones; (**b**) the holding zones at −60 °C; (**c**) the holding zones at 23 °C; (**d**) the holding zones at 60°C; I–VI are numbers of the holding zone at constant temperature; I is 23 °C; II is 60 °C; III is −60 °C; IV is 60 °C; V is −60 °C; VI is 23 °C.

**Table 1 polymers-14-03840-t001:** Options for the ratio of internal and external PC.

Parameter	Designation of the Thicknesses Ratio of the PC						
	*h* _10_90_	*h* _20_80_	*h* _30_70_	*h* _standard_	*h* _70_30_	*h* _80_20_	*h* _90_10_
*h*_1_ (μm)	4.35	8.7	13.05	25	30.45	34.8	39.15
*h*_2_ (μm)	39.15	34.8	30.45	18.5	13.05	8.7	4.35
*h*_1_ (%) of *h*	10	20	30	57.5	70	80	90
*h*_1_ (%) of *h*	90	80	70	42.5	30	20	10

**Table 2 polymers-14-03840-t002:** Thermo-mechanical properties of the model.

Parameter	Mat. 1	Mat. 2	Mat. 3	Mat. 4	Mat. 5	Mat. 6	Mat. 7
*E* (MPa)	67939	49107	65370	72000	1837	7786	68600
*v*	0.168	0.203	0.181	0.170	0.498	0.350	0.340
α·10^−6^ (K^−1^)	1.055	2.675	2.886	0.500	200	50	23

**Table 3 polymers-14-03840-t003:** The fiber birefringence.

Parameter	Analytic	Calculation		
		*R*_coil_ (mm)		
		5	22	50
*B*	4.83 × 10^−4^	4.592 × 10^−4^	4.669 × 10^−4^	4.676 × 10^−4^
Δ (%)	–	4.83	3.24	3.09

## Data Availability

The data presented in this study are available on request from the corresponding author.

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
