# Peer review of "Analysis of the Polymer Two-Layer Protective Coating Impact on Panda-Type Optical Fiber under Bending"

_polymers, 2022, doi:10.3390/polym14183840_

Round 1

Reviewer 1 Report

1. The part about coating in the introduction section deals with a few outdated works of literature. Authors may refer to some of the following recently published articles:

https://doi.org/10.3390/coatings12081213

https://doi.org/10.3390/polym14163304

https://doi.org/10.3390/coatings12081166

https://doi.org/10.3390/coatings12081056

https://doi.org/10.3390/coatings12070989

2. In Chapter 2.2, is there any specific description on the supplier and model of materials?

3. In Chapter 2.4, the reason why this paper only studies the sector of design with a central angle of 0.5° shall be explained in detail. 

4. The author should compare the research content of this paper with the research results of others to show the innovation.

5. The formula used in this paper should give reference standard.

Author Response

Hello.

We made corrections to the article according to the comments and recommendations. Answer to the review in the attached file.

Best wishes, authors

Reviewer 2 Report

In this paper, the authors investigated  on polymer two-layer protective coating and the properties of the Panda type otical fiber during bending with theoretical consideration. Basicallly the work is worthy of publication in the journal. However, English throughout the text should be checked again and polished. In particular the English of the title seems to be aweful. It may be changed as "The Analysis of the polymer two-layer protective coating on properties of the Panda type optical fiber during bending."

Author Response

(The authors gave the same response as above.)

Round 2

Reviewer 1 Report

The paper has been improved. I have no problem.